# PD-L1 Expression in Pituitary Neuroendocrine Tumors/Pituitary Adenomas

**DOI:** 10.3390/cancers15184471

**Published:** 2023-09-08

**Authors:** Giulia Cossu, Stefano La Rosa, Jean Philippe Brouland, Nelly Pitteloud, Ethan Harel, Federico Santoni, Maxime Brunner, Roy Thomas Daniel, Mahmoud Messerer

**Affiliations:** 1Service of Neurosurgery, University Hospital of Lausanne, University of Lausanne, 1005 Lausanne, Switzerland; ethan.harel@chuv.ch (E.H.); roy.daniel@chuv.ch (R.T.D.); mahmoud.messerer@chuv.ch (M.M.); 2Unit of Pathology, Department of Medicine and Technological Innovation, University of Insubria, 21100 Varese, Italy; stefano.larosa@uninsumbria.it; 3Department of Laboratory Medicine and Pathology, Institute of Pathology, University of Lausanne, 1005 Lausanne, Switzerland; jean-philippe.brouland@chuv.ch; 4Department of Endocrinology, University Hospital of Lausanne, University of Lausanne, 1005 Lausanne, Switzerland; nelly.pitteloud@chuv.ch (N.P.); federico.santoni@chuv.ch (F.S.); maxime.brunner@chuv.ch (M.B.)

**Keywords:** biological behavior, immune checkpoint inhibitors, immunotherapy, PD-L1, PitNET, pituitary adenoma, prognosis, somatotroph pituitary tumors, gonadotroph pituitary tumors

## Abstract

**Simple Summary:**

The biological behavior of Pituitary Neuroendocrine Tumors (PitNET) remains unclear. Many efforts have been performed in order to clarify this point. The expression of Programmed cell death ligand 1 (PD-L1) has been associated to a more aggressive behavior in different solid tumors but its impact on PitNET is unclear. Our study analyzes the expression of this protein in a cohort of PitNET and investigates its association with the radiological and pathological behavior of these tumors. Proliferative tumors expressed higher levels of PD-L1 and specific subtypes of PitNET expressing growth hormone also had a higher expression. No association was found between PD-L1 and radiological features of invasion or recurrence. Larger studies are necessary to evaluate if this protein has a real impact on the biological behavior of pituitary tumors and to understand if it can be a useful target for immunotherapy in refractory cases.

**Abstract:**

Background and aim: About a third of Pituitary Neuroendocrine Tumors (PitNETs) may show aggressive behavior. Many efforts have been performed for identifying possible predictive factors to early determine the future behavior of PitNETs. Programmed cell death ligand 1 (PD-L1) expression was associated with a more aggressive biology in different solid tumors, but its role in PitNET is not well-established yet. Our study aims to analyze PD-L1 expression in a surgical cohort of PitNETs to determine its association with radiological invasion and pathology findings, as well as with long-term recurrence rates. Methods: We performed a retrospective analysis in a series of 86 PitNETs. Clinical presentation and radiological features of the preoperative period were collected, as well as pathological data and follow-up data. The rate of PD-L1 expression was immunohistochemically evaluated and expressed as a tumor proportion score (TPS). We assessed its relationship with cavernous sinus invasion and Trouillas’ classification as primary outcomes. Secondary outcomes included the TPS’ relationship with histopathological markers of proliferation, hormonal expression, tumor size and long-term recurrence rates. We calculated the optimal cut-point for the primary outcomes while maximizing the product of the sensitivity and specificity and then we evaluated the significance of secondary outcomes with logistic regression analysis. Results: Eighty-six patients were included in the analysis; 50 cases were non-functional PitNETs. The TPS for PD-L1 showed a highly right-skewed distribution in our sample, as 30.2% of patients scored 0. Using Trouillas’ classification, we found that “proliferative” cases have a significantly higher probability to express PD-L1 in more than 30% of tumor cells (OR: 5.78; CI 95%: 1.80–18.4). This same cut-point was also associated with p53 expression. A positive association was found between PD-L1 expression and GH expression (*p* = 0.001; OR: 5.44; CI 95%: 1.98–14.98), while an inverse relationship was found with FSH/LH expression (*p* = 0.014; OR = 0.27, CI 95%: 0.10–0.76). No association was found with CS invasion, tumor size, bone erosion or dura invasion. We could not find any association between PD-L1 expression and recurrence. Conclusions: PD-L1 expression was associated with proliferative grades of Trouillas’ classification and p53 expression. We also confirmed a higher expression of PD-L1 in somatotroph tumors. Larger studies are necessary to investigate the relationship between PD-L1 expression and aggressive behaviors.

## 1. Introduction

Pituitary neuroendocrine tumors (PitNETs) account for approximately 10% to 15% of intracranial tumors [1,2]. The term ‘PitNET’ was first proposed in 2017 by a group of pathologists to replace the term ‘pituitary adenoma’ to underline the fact that their clinico-biological behavior at diagnosis is often not easily predictable [3,4] and their management is a rapidly evolving field. The term PitNET was then officially introduced in the current WHO classification of pituitary tumors (2022), which is based on a detailed histological subtyping according to the tumor cell lineage, cell type and related characteristics [5].

Most PitNETs are indolent and display noninvasive behaviors, with slow growth confined to the sella or displacing the surrounding tissues. Nevertheless, tumor growth and hormone hypersecretion may cause significant morbidity and mortality in a subset of patients, as up to 35% of PitNETs are invasive, with infiltration of neighboring tissues such as the cavernous sinuses, bone, sphenoid sinus and surrounding nerve sheaths [6]. These PitNETs with aggressive clinical behaviors exhibit a higher risk of local recurrence and invasion of surrounding structures as well as metastatic dissemination in a minority of cases [7,8,9,10,11,12,13], and multiple surgical procedures or extended approaches could be required to address parasellar or subarachnoidal components in symptomatic patients or those with functional PitNET [11]. Radiotherapy can also be indicated in well-selected cases to control tumor growth [14,15] and temozolomide (TMZ) was recommended as a first-line medication for refractory PitNETs by the European Society of Endocrinology [8]. However, only two thirds of patients seem to respond to this medical treatment [16,17,18].

Numerous efforts have been undertaken to identify potential prognostic factors that may indicate a higher likelihood of aggressive behavior [19], as well as to predict a patient’s response to both radiation therapy and medical treatment.

Immune checkpoints, which serve as key regulators of immune activation, play a crucial role in moderating the intensity of immune responses, upholding self-tolerance and minimizing tissue harm. Tumors may take advantage of these immune checkpoints by inhibiting the activation of T cells. The presence of cytotoxic T-lymphocyte-associated protein 4 (CTLA-4) and programmed cell death ligand 1 (PD-L1) in tumor cells may assist tumors in evading the host’s immune system. The expression of PD-L1 has been linked to aggressive behavior in lung tumors [20,21,22] and melanoma [23], but its biological value in PitNET remains unclear. Previous studies described the expression of PD-L1 in somatotroph and lactotroph tumors and in PIT-1 positive plurihormonal PitNETs [24,25]. Its expression is instead rare in transcription factor negative PitNETs, gonadotroph and corticotroph tumors [26].

The aim of our study is to analyze PD-L1 expression in our surgical cohort of PitNETs to evaluate if it correlates with radiological criteria of aggressiveness, namely, with invasion of the cavernous sinus (CS), with histopathological markers of proliferation and with the hormonal profile. We also wanted to analyze the relationship between PD-L1 expression and long-term recurrence.

## 2. Materials and Methods

We analyzed all PitNET cases operated upon at the Neurosurgical Department of the University Hospital of Lausanne, Switzerland, between January 2014 and December 2019. Ethical approval was obtained before starting the study (CER-VD 2020–01338) and it was conducted in accordance with the declaration of Helsinki. Only patients giving their informed consent for this research were included.

Clinical presentation and preoperative radiological features were collected, as well as the pathological data. Postoperative outcomes in terms of extent of resection, complications and rate of recurrence at last follow-up were analyzed. Only patients with a minimal follow-up of 36 months were included.

Magnetic resonance images (MRI) were performed on 1.5 or 3T machines (Siemens, Erlangen, Germany), with 2-mm or 1.5-mm thick slices, respectively, and including a standard protocol with coronal T2-weighted sequences and coronal and sagittal T1-weighted spin-echo sequences, performed before and after injection of gadolinium contrast media. The longest diameter was recorded in mm and the invasion of the cavernous sinus was scored by a neuroradiologist according to the modified Knosp grade [27]. Invasion of the CS was also recorded according to intraoperative findings.

All samples were fixed in buffered formalin and processed to paraffin. We then performed a staining with hematoxylin–eosin for a morphologic evaluation and immunohistochemical (IHC) stains were performed using the Ventana Benchmark XT autostainer (Ventana Medical System, Tucson, AZ, USA) to evaluate hormonal expression.

We classified our cases according to Trouillas’ classification [28]: tumors were considered invasive according to the presence of cavernous sinus invasion, while proliferation was evaluated according to the following histopathological criteria:-Mitotic count greater than 2 mitoses per 10 HPFs-Ki67-labeling index greater or equal to 3%-p53 immunoreactivity in more than 10% of cells

At least 2 out of the 3 criteria should be present to define a tumor as proliferative. Patients were then classified as 1 (non-invasive) versus 2 (invasive), combined with *a* (non-proliferative) or *b* (proliferative). Four categories, 1a, 1b, 2a and 2b were thus defined. Score 3 included PitNETs presenting with metastases.

Continuous variables are presented as mean ± standard deviation (SD), and categorical variables as number and percentage. The rate of PD-L1 expression was evaluated at immunohistochemistry (clone SP263, Ventana platform, Oro Valley, AZ, USA) and expressed as tumor proportion score (TPS) by estimating the percentage of neoplastic cells showing membranous staining. It was assessed by one pathologist blind to the clinical data and previous pathological findings. We considered the TPS as a continuous predictor, and we decided to assess its relationship with radiological features of invasion and with Trouillas’ classification as primary outcomes. Secondary outcomes included the TPS’ relationship with specific histopathological markers of proliferation, hormonal expression and tumoral size as well as with long-term recurrence rates. We assessed the likelihood of each outcome for each five unit increase in the TPS using logistic regression models and checked linearity using the Lowess smoother. We then calculated the optimal cut-point for the primary outcomes while maximizing the product of the sensitivity and specificity [29], and we used TPS as a dichotomic variable. We also evaluated the significance of secondary outcomes with a logistic regression analysis as well as the corresponding OR.

We assessed data distribution using the Kolmogorov–Smirnov test. The significance level was set at a *p*-value < 0.05. Statistical analysis of the data was performed with Stata v17 software (StataCorp, College Station, TX, USA).

## 3. Results

Eighty-six patients were included in the present analysis and the process of patients’ selection is detailed in Figure 1.

Mean age at surgery was 55.2 y (SD +/− 14.7 y), and 40 patients were women (46%). Most PitNETs included in our study were nonfunctioning (50 of 86 patients; 58%); clinical presentation is detailed in Table 1. MacroPitNETs were detected in 90.7% of cases (78/86), with a mean maximal diameter of 21.9 mm +/− 9.9 mm. Eighteen patients had large or giant tumors (5 giant PitNET and 13 large PitNET). Suprasellar extension was observed in 65% of cases (56/86 patients) while an infrasellar extension was only present in one case. CS invasion was classified according to Knosp’s classification [27] (Table 1) and it was confirmed intraoperatively in 51 cases (59.3%; all Knosp 3 and 4 and some grade 2). Bone erosion of the sellar floor was described as an intraoperative finding in 26 out of 81 patients (32%). According to the pathological analysis, dura mater infiltration was analyzed in only 14 cases and an infiltration was detected in 6 of them (42.8%). Hormonal expression at immunohistochemical analysis is detailed in Table 1 according to the 2017 WHO Classification of Pituitary Tumors. Trouillas’ classification was applied on our surgical cohort and the results are reported in Table 1.

The median follow up was of 72 months (IQR: 3.6 mo) and 12 cases experienced a recurrence (14%).

A second surgery was performed in 12 cases (14%).

TPS for PD-L1 showed a highly right-skewed distribution in our sample, with 26/86 of patients (30.2%) scoring zero. Thirty-one patients (36%) presented TPS equal or superior to 30%; 20 (23%) presented TPS equal or superior to 50% and only 10 patients (11.6%) presented TPS equal or superior to 80%. We used a logistic regression model to evaluate the association between TPS as a continuous parameter and the different radiological and histopathological features of the tumors, and then we calculated the optimal cut-point to maximize sensitivity and specificity.

Concerning Trouillas’ classification, when merging the non-proliferative cases (1a + 2a) versus the proliferative ones (1b + 2b), we found that the optimal cut-point for TPS was 30% (*p* = 0.004). Thus, proliferative cases have a significantly higher probability to express PD-L1 in more than 30% of tumoral cells (OR: 5.78; CI 95%: 1.80–18.4; NPV: 91.2%) (Figure 2).

Using this same cut-point, we found a significant association between TPS and p53 nuclear expression in more than 10% of cells, but not with the number of mitosis or Ki67 positive cells or with dura mater infiltration (Figure 2).

We did not find an association between TPS and radiological features of invasion at that stage, such as cavernous sinus invasion, Knosp grade, tumor size or intraoperative findings of bone erosion (Figure 3).

Concerning hormonal expression, using the same cut-point of TPS at 30%, a positive association was found with GH expression (*p* = 0.001; OR: 5.44; CI 95%: 1.98–14.98), and this association was also confirmed clinically, as patients with signs and symptoms of acromegaly presented the higher TPS values (*p* < 0.0001). An inverse relationship was found with FSH/LH expression, as low levels of PD-L1 expression (TPS < 30%) were associated with gonadotrophin expression (*p* = 0.014; OR = 0.27, CI95%: 0.10–0.76) (Figure 4).

No association was found with the expression of other hormones, nor with long-term recurrence (OR: 0.65; 95% CI: 0.16–2.63; p: 0.55).

## 4. Discussion

The upregulation of PD-L1 may represent a pathway to evade immune surveillance and it could thus be associated with a more aggressive behavior in PitNET, through the suppression of the host immune response [30,31,32]. Some studies reported significant association between high PD-L1 expression and functioning PitNET and a high Ki-67 proliferative index [24,25,33], while Sato et al. described an association between CS invasion and high PD-L1 expression [34]. However, these findings remain under debate, as other authors reported that PD-L1 expression was significantly lower in invasive non-functional PitNET [35].

In our surgical series, the optimal TPS cut-point able to differentiate proliferative from non-proliferative cases, according to Trouillas’ classification, was established at 30%. Although, PD-L1 expression did not collate with mitotic activity of Ki-67 index, a TPS > 30% was observed in p53 positive cases (p53 expression in more than 10% of cells). We could not find any association with CS invasion, Knosp grade or tumor size. Concerning hormonal immunohistochemical analysis, TPS values > 30% were associated with GH expression, while an inverse relationship was found with gonadotroph tumors. These findings are in line with previously published studies [24,26] and this could suggest a possible association between the expression of PD-L1 and the transcriptional factor PIT-1 [26]. No relationship between TPS and recurrence rate was found in our study and thus we could not confirm the findings of Shi et al. [36]. However, the authors concentrated on PD-L1 expression in the pediatric and adolescent population, showing that significantly higher level of PD-L1 were expressed in the pediatric and adolescent group when compared to older patients [36].

PD-L1 expression may represent an interesting target for immunotherapy in patients with functioning aggressive PitNET not responding to conventional treatment, considering that 33% of patients receiving TMZ experience disease progression [13,16,17]. Immune checkpoint inhibitors have gained significant attention as a cancer treatment strategy in recent years as they have demonstrated remarkable efficacy in achieving prolonged tumor remission in various solid malignancies [37,38,39,40,41]. However, the effectiveness of immunotherapy remains limited for certain tumors, resulting in low response rates [37,41]. As a result, there is a pressing need for biomarkers that can differentiate sensitive patients and predict treatment outcomes. The blockade of the programmed cell death 1 (PD-1) or of its ligand (PD-L1) may activate T cell stimulatory signaling, thereby enhancing antitumor T cell cytotoxicity and promoting tumor destruction [42,43,44,45,46] (Figure 5).

An accurate identification of the subgroups of patients with PitNET who may benefit from this immunotherapy is the first critical step in the risk-benefit assessment of these treatments, as patients treated with checkpoint inhibitors can develop severe inflammatory diseases, and the efficacy of these treatments is not yet established [47,48,49]. The most commonly used diagnostic test for checkpoint inhibitors is immunohistochemistry (IHC) for PD-L1 on the target cells, supposing that tumors with high levels of PD-L1 expression are most likely to respond [46,50,51]. TPS corresponds to the percentage of neoplastic cells showing membranous staining on three sides of the cell and a TPS > 50% is used to suggest a likely net benefit for checkpoint inhibitors [51]. However, its application in PitNETs is still limited [52] and a cut-off to establish the usefulness of these treatments has not been established yet. Beside PD-L1 expression, features of the tumor microenvironment may impact on the efficacy of immune checkpoint inhibitors, namely tumor-infiltrating T-lymphocytes and sequencing-based mutations [53,54].

Although only a few data points are available, murine and cell line studies indicate that PD-L1 is a potential target in PitNET [30]. The efficacy of anti-PD-L1 antibodies in PitNET has been examined in both subcutaneous and intracranial murine models of Cushing disease, showing an efficacy in reducing tumor mass and hormone secretion and in blocking tumor proliferation [33]. To date, there are two registered clinical trials of immunotherapy for pituitary tumors: one phase II clinical trial of nivolumab combined with ipilimumab for patients with aggressive pituitary tumors (NCT04042753) and another phase II clinical trial testing the combination of nivolumab and ipilimumab for rare tumors, including pituitary tumors (NCT02834013).

Furthermore, immune-checkpoint inhibitors can be useful in the treatment of ACTH-secreting metastatic PitNET [52,55] but no association between PD-L1 expression and clinical response was found [55]. This could be attributed to previous treatments with TMZ, that can induce alterations in the mismatch repair system and enhance the efficacy of treatment with checkpoint inhibitors [56]. Furthermore, recurrent tumors seem to exhibit lower levels of PD-L1 than primary PitNET [10].

Further research is warranted to identify robust biomarkers that can accurately predict the response to immunotherapy. Although several studies have examined the potential of measuring PD-L1 expression as a predictor of response to checkpoint inhibitors, its clinical utility is showing its limitations. PD-L1 staining has low prediction accuracy and its levels are dynamic, rendering it an unreliable tool for patient selection in the PD-1/PD-L1 pathway blockade. Hence, there is an urgent need to develop more reliable and precise biomarkers for predicting the response to immunotherapy in clinical practice [57].

### Limitations

The retrospective observational design of this study has certain methodological drawbacks that are commonly associated with retrospective data, which limits its scope. Moreover, the comparative analysis of subgroups should be approached with caution, as the limited number of patients in the study could affect the power of the analysis. Only patients agreeing to the research were included and this could also introduce a supplementary selection bias. Therefore, it is necessary to conduct further clinical studies with larger patient cohorts to validate these findings.

## 5. Conclusions

In our surgical cohort, PD-L1 expression was associated with proliferative grades according to Trouillas’ classification and p53 expression. We were unable to find an association between PD-L1 expression and cavernous sinus invasion or Ki-67 expression. We also confirmed a positive association between PD-L1 expression and somatotroph PitNET and a negative association with gonadotroph tumors. Larger studies are necessary to investigate the relationship between PD-L1 expression and the biological behavior of PitNET.

## Figures and Tables

**Figure 1 cancers-15-04471-f001:**
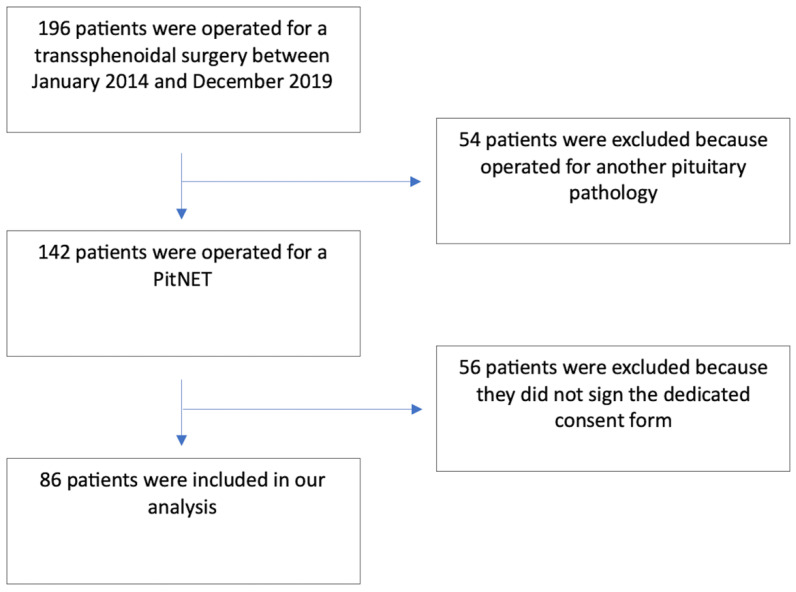
Flow-chart showing the process of patients’ selection and inclusion in our analysis.

**Figure 2 cancers-15-04471-f002:**
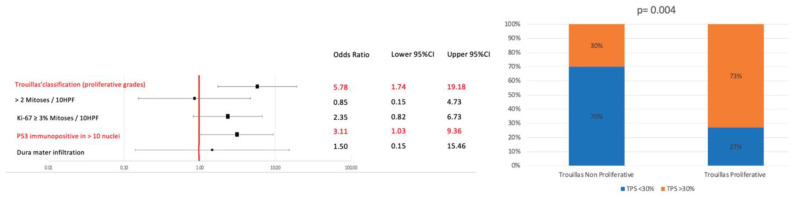
A higher PD-L1 expression was associated with proliferative grades of Trouillas’ classification along with a higher expression of P53 (more than 10 cells with a strong nuclear staining per 10 HPF). No association was found with other proliferative markers, namely Ki67 labeling index or mitotic count (**left panel**). Considering a cut-off of 30% of cells expressing PD-L1, a very strong association was found with proliferative Trouillas’ grades (*p* = 0.004; **right panel**).

**Figure 3 cancers-15-04471-f003:**
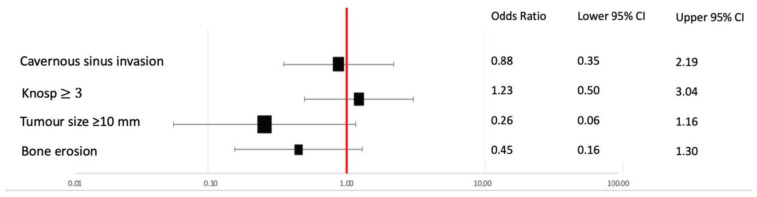
No relationship was found between PD-L1 expression and radiological or pathological features of invasion. Bone erosion was evaluated on preoperative CT and during surgery, while dural infiltration was evaluated through histopathological analysis.

**Figure 4 cancers-15-04471-f004:**
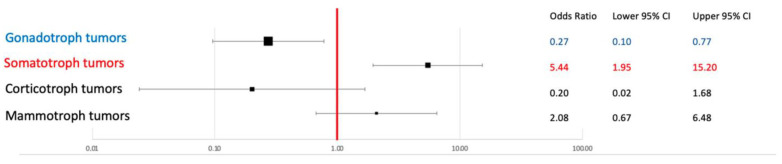
Concerning the hormonal expression at immunohistochemistry, a strong positive association was found between PD-L1 expression and GH expression (somatotroph tumors), while an inverse relationship was found with FSH/LH expression, as gonadotroph tumors were associated with a very low expression of PD-L1. No relationship was found between ACTH- and PRL-expressing tumors and PD-L1 expression.

**Figure 5 cancers-15-04471-f005:**
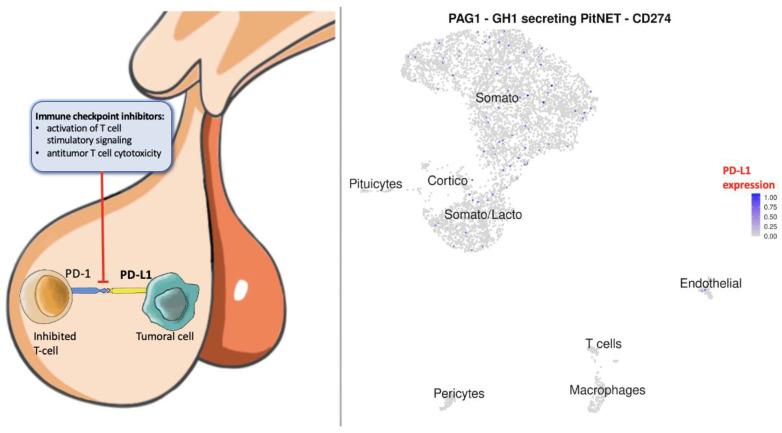
On the left, the role of PD-L1 (programmed-death ligand 1) in tumoral cells is shown. It binds to PD-1 expressed by T-cells to inhibit their functions. The tumor can thus evade host immunity and continue to proliferate. The blockade of this cascade (PD-L1) allows the activation of T cell stimulatory signaling, thereby enhancing antitumor T cell cytotoxicity and proinflammatory cytokine production, promoting tumor destruction. On the right, we provided a UMAP plot of single-cell RNA sequencing data showing PD-L1 normalized expression (CD274) in a somatotroph PitNET.

**Table 1 cancers-15-04471-t001:** Epidemiological, clinical, radiological and pathological data are here summarized.

	N° Patients (%)
Epidemiology: Mean age: 55.2 y (SD +/− 14.7 y) Women	40 (46%)
Clinical presentation * Acromegaly signs/symptoms Hyperprolactinemia signs/symptoms Cushing disease Non-functioning PitNET ** Apoplexy	20/86 (23%) 10/86 (11.6%) 8/86 (9.3%) 50/86 (58%) 4 (4.6%)
Vision: Visual field impairment Visual acuity impairment Palsy of oculomotor nerves in the CS	25/86 (29%) 10/86 (11.6%) 6 (7%)
Preoperative endocrine deficit: Panhypopituitarism hypogonadism hypocorticism hypothyroidism diabetes insipidus stalk effect	9/86 (10.5%) 40/83 (48.2%) 13/85 (15.3%) 18/84 (21.4%) 0 (0%) 28/79 (35.4%)
Knosp grade: Knosp 0 Knosp 1 Knosp 2 Knosp 3a Knosp 3b Knosp 4	7 (8.1%) 9 (10.5%) 30 (34.9%) 22 (25.6%) 7 (8.1%) 11 (12.8%)
PitNET classification: Somatotroph tumors Lactotroph tumors Mammosomatotroph tumors Corticotroph tumors Gonadotroph tumors Plurihormonal tumors	14 (16.3%) 9 (10.5%) 4 (4.6%) 10 (11.6%) 42 (48.8%) 7 (8.2%)
Trouillas’ classification: Grade 1a Grade 1b Grade 2a Grade 2b Grade 3	35/85 (41.15%) 4 (4.7%) 35 (41.15%) 9 (10.6%) 2 (2.4%)

* Two patients presented signs and symptoms of hypersecretion of both GH and PRL. ** surgical indication for growing tumors or visual impairments. Abbreviations: CS: cavernous sinus; PitNET: pituitary neuroendocrine tumor.

## Data Availability

The data can be shared up on request.

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
