# Peer review of "PD-L1 Expression in Pituitary Neuroendocrine Tumors/Pituitary Adenomas"

_cancers, 2023, doi:10.3390/cancers15184471_

Round 1
Reviewer 1 Report
- Was there any pattern in the specific proliferation markers that did (p53) or did not (Ki-67, mitoses) correlate with PD-L1 levels? Could there be something unique about p53 being associated while other markers were not?
- Were there any associations found between PD-L1 levels and specific presenting symptoms or hormone hypersecretion profiles?
- Can you speculate on any biological or mechanistic reasons for the differences in PD-L1 expression seen between somatotroph vs gonadotroph subtypes?
- Was there any pattern in the specific proliferation markers that did (p53) or did not (Ki-67, mitoses) correlate with PD-L1 levels? Could there be something unique about p53 being associated while other markers were not?
The English language quality of this research article is quite good overall. The writing is clear and comprehensible, with relatively few grammatical errors or awkward phrasing. Some sentences could be tightened up or smoothed out, but the language does not impede understanding of the science being presented.
Author Response
Reviewer 1
- Was there any pattern in the specific proliferation markers that did (p53) or did not (Ki-67, mitoses) correlate with PD-L1 levels? Could there be something unique about p53 being associated while other markers were not?
This is a very interesting topic. The debate is open concerning the impact of p53 and of Ki-67 in the prediction of the biological behavior of pituitary neuro-endocrine tumors.
Questions rise about the usefulness of these markers in PitNET and in the new classification updated in 2022 pathologists chose to classify PitNET according to the transcriptional factor and not according to the histo-pathological markers of proliferations. To our knowledge, up to now, no study investigated the specific patterns of these markers in PitNET. Our work showed that there was a correlation only with p53 but we did not want to advance hypothesis on this subject and the literature is scarce.
- Were there any associations found between PD-L1 levels and specific presenting symptoms or hormone hypersecretion profiles?
We found a correlation between high PD-L1 expression and acromegaly manifestations.
This information was added in the Results section as follows:
Concerning hormonal expression, using the same cut-point of TPS at 30%, a positive association was found with GH expression (p=0.001; OR: 5.44; CI 95%: 1.98-14.98), and this association was also confirmed clinically, as patients with signs and symptoms of acromegaly presented the higher TPS values (p<0.0001).
- Can you speculate on any biological or mechanistic reasons for the differences in PD-L1 expression seen between somatotroph vs gonadotroph subtypes?
We do not know if some molecular cascades are shared between hormonal expression and PD-L1 expression. It could be interesting to analyze if PD-L1 expression was higher in sparsely or densely granulated tumors expressing GH and this could be the topic of a further paper.
Furthermore, some pathways could be shared between the expression of the transcriptional factor PIT-1 and PD-L1. This hypothesis was advanced by Turchini et al. in 2021 and we are further studying this hypothesis in an ongoing paper.
We added this information in the discussion section:
Concerning hormonal immunohistochemical analysis, TPS values > 30% were associated with GH expression, while an inverse relationship was found with gonadotroph tumors. These findings are in line with previously published studies. [23, 25] and this could suggest a possible association between the expression of PD-L1 and the transcriptional factor PIT-1.
The English language quality of this research article is quite good overall. The writing is clear and comprehensible, with relatively few grammatical errors or awkward phrasing. Some sentences could be tightened up or smoothed out, but the language does not impede understanding of the science being presented.
We would like to thank the reviewer. We corrected the English of the paper.
Reviewer 2 Report
Although there are aspects of the research article that can be commended, in this review emphasis will be on limitations and necessity to address problems.
Line 2. I suggest to use "association" instead of "correlation" in the title.
Lines 32 and 130 mention performed ROC analyses, yet there is nothing about ROC in the results or discussion (I found no evidence that article does have supplementary or annex data, which if there is one I have missed)
Lines 85-86 Define radiological criteria of aggressiveness as invasion of CS, yet data show that there is no association (Fig 2), therefore making claim in the title either misleading or at least confusing. I suppose the title was meant to reflect association with Trouilla's classification (Fig 3), but article does not mention "aggressiveness" when describing Trouilla's classification.
Material and methods: Threshold for statistical significance is not mentioned.
Line 116 states, that quantitative values are expressed as mean and SD yet line 155 are about median and IQR. Also test for normal distribution for quantitative variables is not mentioned (likely performed, otherwise there would not be median and IQR for likely non-normally distributed data).
Figure 1. High level of non-consenting patients suggest true exercise of patient's free will (and high research ethics), but also could introduce biases, especially as all patients seem to have agreed for long term follow up (Lines 97 and 98 does not exclude any additional patients according to Figure 1). Therefore, this should be included in limitations.
Table 1 describes patients in many facets, but the newest and currently quite important is missing. PitNET lineage marker analysis, even if not available for all patients should be included.
Table 1 "Clinical presentation" section adds up to 88/86 [likely] due to Mammosomatotroph tumors, but could be confusing and would be better to mention this in table's footnotes.
Table 1 "PitNET classification" section Mammosomatotroph tumours are also plurihormonal, should be elaborated about this division
Table 1 "Trouillas’ classification" section shows 85 tumours graded. What happened to 86th?
Lines 158-160 Description of "TPS for PD-L1" distribution is unclear. 26 have 0, 20 >50% and rest (40) I assume are between 0 and 50 but not totally clear. Also this part would benefit from reporting how many are above and below TPS for PD-L1 cut off point (30%).
Lines 161-162. I think wording "TPS as a continuous parameter" is wrong. TPS values were transformed into categorical classes at cut-off point 30% and then used for association analyses with logistical regression. Right? Otherwise you have used some exotic statistics where continuous variable allows to calculate odds without transformation. Statistics description parts should be clarified.
Lines 167-170 a) There seems to be missing information for figure 2 description
b) main Results paragraph is in font of figure footnotes
c)1A, 1B and 2A, 2B is in uppercase although line 114 denominates these classes with lowercase letter
Figures 2-3 Why there is evasion to use TPS for PD-L1 as continuous parameter and use box plots to visualize comparisons of medians and quartiles between categorical states?
Figure 4 a) Would also benefit from correlating quantitative values of TPS for PD-L1 and hormone expression.
b) Otherwise hormone expression cut-offs/classes is not described.
c)IHC (I presume) of hormone expression is not described in Material and methods. I would suggest using "hormone production" as not to confuse with expression of hormone genes or clarify if it means gene expression.
Figure 5 is not necessary. Left side completely necessary. Right side is out of context, I presume it is your own data of of single-cell RNA sequencing study but if its not analysed in this research article, then it is out of place as those are likely unpublished and not peer reviewed data.
Terms "association" and "correlation" are not complete synonyms and should not be used interchangeably freely. Consider their proper use in all instances of the manuscript.
Regards
Author Response
Reviewer 2
Although there are aspects of the research article that can be commended, in this review emphasis will be on limitations and necessity to address problems.
Line 2. I suggest to use "association" instead of "correlation" in the title.
We corrected the title according to this suggestion.
Lines 32 and 130 mention performed ROC analyses, yet there is nothing about ROC in the results or discussion (I found no evidence that article does have supplementary or annex data, which if there is one I have missed)
You are right. We performed ROC analysis but we decided not to report these results as they were redundant with other analyses. This sentence was deleted in the text.
Lines 85-86 Define radiological criteria of aggressiveness as invasion of CS, yet data show that there is no association (Fig 2), therefore making claim in the title either misleading or at least confusing. I suppose the title was meant to reflect association with Trouilla's classification (Fig 3), but article does not mention "aggressiveness" when describing Trouilla's classification.
The reviewer rises a very important debate. The definition of aggressiveness for PitNET remains unclear as many criteria were proposed but are still insufficient.
Despite numerous studies and advances in prognostic classifications, no single morphological or histological markers has been shown as yet to reliably predict aggressive behavior. No new PitNET grading system was introduced in the WHO 2022 classification and at present identification of aggressive tumors should be made on individual basis, considering the histological subtype, proliferative potential and tumor invasion assessment, as well described by Melmed S. et al in 2022.
Thank to this remark we decided to change the title of the manuscript and to avoid the term “aggressiveness”, to avoid any misunderstanding. This important reference was also added in the text.
Material and methods: Threshold for statistical significance is not mentioned.
The significance level was set at a p-value <0.05.
This information was added at the end of the Material and methods section.
Line 116 states, that quantitative values are expressed as mean and SD yet line 155 are about median and IQR. Also test for normal distribution for quantitative variables is not mentioned (likely performed, otherwise there would not be median and IQR for likely non-normally distributed data).
We assessed data distribution using the Kolmogorov-Smirnov test. This information was added at the end of the Material and methods section.
Figure 1. High level of non-consenting patients suggests true exercise of patient's free will (and high research ethics), but also could introduce biases, especially as all patients seem to have agreed for long term follow up (Lines 97 and 98 does not exclude any additional patients according to Figure 1). Therefore, this should be included in limitations.
We thank the reviewer for this remark and we added the following sentence to the Limitation section:
The retrospective observational design of this study has certain methodological drawbacks that are commonly associated with retrospective data, which limits its scope. Moreover, the comparative analysis of subgroups should be approached with caution, as the limited number of patients in the study could affect the power of the analysis. Only patients agreeing to the research were included and this could also introduce a supplementary selection bias.
Table 1 describes patients in many facets, but the newest and currently quite important is missing. PitNET lineage marker analysis, even if not available for all patients should be included.
According to our ethical protocol, we reported our cohort patients according to the 2017 classification, as we used this classification during the performance of this study.
Nevertheless, your remark is relevant and we are prospectively analyzing PD-L1 expression in our surgical cohort of PitNET according to transcriptional factors analysis (WHO 2022 classification) and we are planning to report these results in a further paper.
Table 1 "Clinical presentation" section adds up to 88/86 [likely] due to Mammosomatotroph tumors, but could be confusing and would be better to mention this in table's footnotes.
This important information was added in the footnotes of Table 1.
Table 1 "PitNET classification" section Mammosomatotroph tumours are also plurihormonal, should be elaborated about this division
Mammosomatotroph are plurihormonal tumors by definition but they represented separate entities according to the WHO 2017 classification. This was also confirmed with the WHO 2022 classification. In fact, under the term plurihormonal we include adenomas with unusual immunohistochemical combinations (See Ref Mete O et al. 2017, DOI 10.1007/s12022-017-9498-z and Asa S. et al. 2022, https://doi.org/10.1007/s12022-022-09703-7).
Table 1 "Trouillas’ classification" section shows 85 tumours graded. What happened to 86th?
Pathological evaluation was not complete and the histo-pathological markers of proliferation were not fully assessed. Thus, we decided to exclude this patient.
Lines 158-160 Description of "TPS for PD-L1" distribution is unclear. 26 have 0, 20 >50% and rest (40) I assume are between 0 and 50 but not totally clear. Also this part would benefit from reporting how many are above and below TPS for PD-L1 cut off point (30%).
We added this information in the Results section as follows:
TPS for PD-L1 showed a highly right-skewed distribution in our sample, with 26/86 of patients (30.2%) scoring zero. Thirty-one patients (36%) presented TPS equal or superior to 30%; 20 (23%) presented TPS equal or superior to 50% and only 10 patients (11.6%) presented TPS equal or superior to 80%.
Lines 161-162. I think wording "TPS as a continuous parameter" is wrong. TPS values were transformed into categorical classes at cut-off point 30% and then used for association analyses with logistical regression. Right? Otherwise you have used some exotic statistics where continuous variable allows to calculate odds without transformation. Statistics description parts should be clarified.
In this study, we considered one continuous exposure variable, TPS, and several binary outcome variables. In logistic regression, the outcome must be binary but the exposure can be continuous. As explained on page 3, line 145, for some analyses we modelled the odds of cancer invasion or proliferation (yes/no outcome) for each 5-unit increase in TPS (continuous). Once we found the optimal cut-off point for TPS using the Liu method, we treated that exposure variable as a binary variable in subsequent analyses (<30 vs 30+).
We modified the description of our statistical analysis in the Material and Methods section as follows:
We considered TPS as a continuous predictor and we decided to assess its relationship with radiological features of invasion and with Trouillas’ classification as primary outcomes. Secondary outcomes included TPS relationship with specific histopathological markers of proliferation, hormonal expression, tumoral size as well as with long-term recurrence rates. We assessed the likelihood of each outcome for each five unit increase in TPS using logistic regression models and checked linearity using the Lowess smother. We then calculated the optimal cut point for the primary outcomes while maximizing the product of the sensitivity and specificity[29] and we used TPS as a dichotomic variable. We also evaluated the significance of secondary outcomes with a logistic regression analysis as well as the corresponding OR.
Lines 167-170 a) There seems to be missing information for figure 2 description
- b) main Results paragraph is in font of figure footnotes
For the point a and b there was a problem in text editing. A paragraph of the Results section was missing as well as a part of the legend for Figure 2.
We corrected it. Main text:
Concerning Trouillas’ classification, when merging the non-proliferative cases (1a + 2a) versus the proliferative ones (1b + 2b), we could find that the optimal cut point for TPS was 30% (p=0.004). Thus, proliferative cases have a significantly higher probability to express PD-L1 in more than 30% of tumoral cells (OR: 5.78; CI 95%: 1.80-18.4; NPV: 91.2%) (Figure 2).
Using this same cut point, we found a significant association between TPS and p53 nuclear expression in more than 10% of cells, but not with the number of mitosis or Ki67 positive cells or with dura mater infiltration (Figure 2).
We did not find association between TPS and radiological features of invasion at that stage, such as cavernous sinus invasion, Knosp grade, tumor size or intraoperative findings of bone erosion (Figure 3).
Concerning hormonal expression, using the same cut-point of TPS at 30%, a positive association was found with GH expression (p=0.001; OR: 5.44; CI 95%: 1.98-14.98), and this association was also confirmed clinically, as patients with signs and symptoms of acromegaly presented the higher TPS values (p<0.0001). An inverse relationship was found with FSH/LH expression, as low levels of PD-L1 expression (TPS < 30%) were associated with gonadotrophin expression (p=0.014; OR=0.27, CI95%: 0.10-0.76) (Figure 4).
c)1A, 1B and 2A, 2B is in uppercase although line 114 denominates these classes with lowercase letter
We corrected the text. Thank you for the comment.
Figures 2-3 Why there is evasion to use TPS for PD-L1 as continuous parameter and use box plots to visualize comparisons of medians and quartiles between categorical states?
As we specified in a previous answer, for some analyses we modelled the odds of cancer invasion or proliferation (yes/no outcome) for each 5-unit increase in TPS (continuous). Once we found the optimal cut-off point for TPS using the Liu method, we treated that exposure variable as a binary variable in subsequent analyses (<30 vs 30+). These OR were then represented as box plots.
Figure 4 a) Would also benefit from correlating quantitative values of TPS for PD-L1 and hormone expression. b) Otherwise hormone expression cut-offs/classes is not described.
Unfortunately this analysis was not possible as our pathologists don’t routinely perform quantitative evaluation of hormonal expression.
c)IHC (I presume) of hormone expression is not described in Material and methods. I would suggest using "hormone production" as not to confuse with expression of hormone genes or clarify if it means gene expression.
The following paragraph was added in the Material and Methods section:
All samples were fixed in buffered formalin and processed to paraffin. We then performed a staining with hematoxylin–eosin for a morphologic evaluation and immunohistochemical (IHC) stains were performed using the Ventana Benchmark XT autostainer (Ventana Medical System, Tucson, AZ, USA) to evaluate hormonal expression.
With immunohistochemistry we evaluate a cellular expression of different hormones but this does not correspond to a hormonal production as some tumors may present a positive IHC (hormonal expression) but they are clinically silent as no hormone is released in the bloodstream.
Figure 5 is not necessary. Left side completely necessary. Right side is out of context, I presume it is your own data of of single-cell RNA sequencing study but if its not analysed in this research article, then it is out of place as those are likely unpublished and not peer reviewed data.
This is correct, the right side represents our own data and it allowed us to understand PD-L1 expression at a single-cell level. We wanted to show to the reader to what it corresponds to a molecular level.
Terms "association" and "correlation" are not complete synonyms and should not be used interchangeably freely. Consider their proper use in all instances of the manuscript.
We checked this issue throughout the manuscript and we privileged the term association.
Round 2
Reviewer 1 Report
The manuscript looks better now. Thanks for the clarifications.
Please reformat Table 1. The bullets in the table make it confusing.
NA
Reviewer 2 Report
Thank you for corrections and explanations